# Inactivating hepatitis C virus in donor lungs using light therapies during normothermic ex vivo lung perfusion

Marcos Galasso[1], Jordan J. Feld[2], Yui Watanabe[1], Mauricio Pipkin[1], Cara Summers[1], Aadil Ali[1], Robert Qaqish[1], Manyin Chen[1], Rafaela V.P. Ribeiro[1], Khaled Ramadan[1], Layla Pires[1], Vanderlei S. Bagnato[3], Cristina Kurachi[3], Vera Cherepanov[2], Gray Moonen[1], Anajara Gazzalle[1], Thomas K. Waddell[1], Mingyao Liu[1], Shaf Keshavjee[1], Brian C. Wilson[4], Atul Humar[5] & Marcelo Cypel[1,5]

Availability of organs is a limiting factor for lung transplantation, leading to substantial mortality rates on the wait list. Use of organs from donors with transmissible viral infections, such as hepatitis C virus (HCV), would increase organ donation, but these organs are generally not offered for transplantation due to a high risk of transmission. Here, we develop a method for treatment of HCV-infected human donor lungs that prevents HCV transmission. Physical viral clearance in combination with germicidal light-based therapies during normothermic ex-vivo Lung Perfusion (EVLP), a method for assessment and treatment of injured donor lungs, inactivates HCV virus in a short period of time. Such treatment is shown to be safe using a large animal EVLP-to-lung transplantation model. This strategy of treating viral infection in a donor organ during preservation could significantly increase the availability of organs for transplantation and encourages further clinical development.

[1] Latner Thoracic Surgery Research Laboratories, Toronto General Research Institute, University Health Network, Toronto M5G 2C4 ON, Canada. [2] Toronto Centre for Liver Disease, University Health Network, Toronto General Hospital, Toronto M5G 2C4 ON, Canada. [3] São Carlos Institute of Physics, University of São Paulo Brazil, São Paulo 13566-590, Brazil. [4] Princess Margaret Cancer Centre/Department of Medical Biophysics, University of Toronto, Toronto M5G 2C4, Canada. [5] Multi-Organ Transplant Program, University Health Network, Toronto M5G 2C4 ON, Canada. Correspondence and requests for materials should be addressed to M.C. (email: marcelo.cypel@uhn.ca) or to J.J.F. (email: Jordan.feld@uhn.ca)

Lung transplantation (LTx) is a successful therapy for end-stage lung diseases but the number of available donors do not meet the demand[1,2], partially due to the fact that only 15% of lungs from available donors are used[3]. The shortage of suitable organs leads to high waiting-list mortality rates[4]. Consequently, approaches to increase organ availability are critical for realizing the maximum potential benefit of transplantation[5].

While many strategies continue to be explored to treat common donor lung injuries such as aspiration pneumonia, pulmonary edema, pulmonary emboli and bacterial infection[5–8], no attempts at treating common donor chronic viral infections such as Hepatitis C have been evaluated. HCV affects 2% of North Americans[9–12]. However, in the context of organ donation and the current epidemic of overdose-deaths (OD) in the donor population, some geographic areas in USA report up to 20% of all organ donors being nucleic acid test positive (NAT+) for HCV[13]. Underuse of these organs is particularly relevant, given that these donors are often young, with fewer comorbidities than other donors[14].

To date, lungs from donors testing NAT+ for HCV are generally not used for transplantation due to a virtual 100% risk of transmission to recipients[15]. Moreover, historical data have shown that HCV-negative recipients receiving lungs from HCV-positive donors have significantly worse post-transplant outcomes[12,15]. Thus, if HCV-positive donors could be safely added to the donor pool, it is estimated that at least 1,000 new donors for LTx would be available every year in North America alone[15,16]. While the administration of novel direct-acting antivirals (DAAs) for HCV to patients post-transplant is being assessed as a method of utilizing these organs[17], a more attractive alternative is clearance or inactivation of the virus within the organ prior to transplantation, thereby not only increasing the societal acceptance of using these organs but also avoiding costs, toxicity and drug interactions related to antiviral medication.

The normothermic ex-vivo lung perfusion system (EVLP) is an innovative method to assess lung function in the clinical setting prior to lung transplantation[3]. It also provides an opportunity for individualized therapy for otherwise damaged donor lungs. EVLP has played a pivotal role in the expansion of the donor organ pool[1,3,18]. The benefits of normothermic ex-vivo organ perfusion have also been recently demonstrated in a seminal clinical trial in liver transplantation[19].

Non-pharmacologic approaches such as light-based therapies (LbT) have been historically used to treat blood components. Ultraviolet C (UVC) irradiation, especially in the germicidal spectrum range of 250–270 nm, has shown to be a very effective sterilizing approach[20,21]. Recent data describe HCV inactivation in culture media and in human serum including sterilization of pre-donation blood components[22]. Similarly, photodynamic therapy (PDT), using methylene blue activated with red light irradiation, is another proven method for pre-donation sterilization of blood components in blood banks[23–26]. PDT involves administration of a photosensitizing drug which requires oxygen and light irradiation at a specific wavelength band for activation, thereby forming reactive oxygen species (ROS) and causing irreversible photo-damage to viruses, including HCV and HIV-1[24,25].

Since there is no robust evidence for HCV infecting or replicating inside lung cells, we hypothesized that intravascular and lung tissue mechanical perfusion during normothermic EVLP could be an effective platform to decrease the HCV viral load in donor lungs. The effect of EVLP in decreasing HCV levels in donor organs before transplant has previously been shown in a case report by our group, where an 80% decrease in HCV RNA was seen in the lung tissue after 6 h of EVLP. However, this was not sufficient to prevent recipient infection[26], indicating that

adjuvant strategies to treat HCV during EVLP are required. However, DAAs are unlikely to have a significant effect when treating a lung ex-vivo, since viral replication is thought to not be occurring within the lung itself. We thus investigated the use of UVC and PDT during normothermic EVLP with the goal of treating and preventing Hepatitis C transmission from infected donor organs. Herein, we demonstrate that HCV is inactivated in short period of time using an innovative combination of normothermic ex vivo perfusion and LbT using infected human lungs. Furthermore, the safety of the approach is confirmed in a pig LTx model.

## Results

**The illumination device.** We first developed a specific light delivery system suitable for use during normothermic EVLP. Given the photon energies involved, UVC and PDT have similar virucidal effects, primarily degradation of viral genomic material and amino acids[27], leading to inactivation[20]. A customized apparatus (U.S. Provisional Patent Application No. 62/481,523 (Y/R 2016–080–01; O/R 10723-P52847US00) was designed for this study (Fig. 1a). It was conceived to be part of the clinical EVLP circuitry (Fig. 1b, Supplementary Movie 1), and contains two main components: an inner hollow quartz tube and a polyvinyl chloride (PVC) outer tube that seals the device and prevents light leakage. A UVC lamp (OSRAM Puritec® HNS 4 W G5, 254 nm) or customized LED light source (centered emission at 660 nm) are placed in the inner of the quartz tube, depending on the treatment. The measured irradiances at the quartz tube surface of the UVC and PDT were of 31 mW/cm² and 20 mW/cm², respectively. During organ perfusion, the perfusate is circulated through the organ and continuously irradiated by light in the circuit, in a cumulative manner. Since human albumin is a key component in the EVLP perfusate (Steen® solution, XVIVO Perfusion, CO, USA), we evaluated the albumin molecular transformation after UVC irradiation by gel electrophoresis. No albumin fragmentation or polymerization was seen with up to 6 h of UVC. Albumin concentration levels were in fact maintained after 12 h of irradiation, as analyzed spectrographically (Supplementary Figs. 1a, b).

**Effect of EVLP and LbT on HCV NAT+ human donor lungs.** To evaluate the effect of EVLP as an isolated strategy and also as a platform for LbT against HCV in the lung tissue and perfusate solution, we studied 9 clinically rejected pairs of human lungs from NAT+ HCV donors. All donors had a signed research consent declaration. The lungs were recovered using standard preservation protocols and flushed with cold low-potassium dextran solution (Perfadex®, XVIVO Perfusion, CO, USA) and preserved at 4 °C for transport. Upon arrival at the hospital, each lung was placed in a separate clinical-grade normothermic EVLP circuit for 9 h using Steen solution. One lung was randomly selected as the control group (standard EVLP protocol; $n = 9$) and the other lung was assigned to different treatment conditions ($n = 3$ each) (Study design, Fig. 2a): (1) Circuit exchange (complete replacement of perfusate solution and circuit after 3 h of EVLP); (2) UVC treatment (254 nm, 31 mW/cm², irradiation time 8 h) applied to the perfusate (flow 1 L/min); (3) PDT using methylene blue (MB, Sigma Aldrich, MI, USA) diluted in the perfusion solution (1 μmol/L) and activated with red light (660 nm, 20 mW/cm², irradiation time 8 h) applied to perfusate (flow 1 L/min). The advantage of this design is that both control and treated lungs came from the same donor and therefore had the same baseline viral load.

The initial standard flush preservation of the lungs prior to starting EVLP resulted in an average of 1-log reduction of HCV

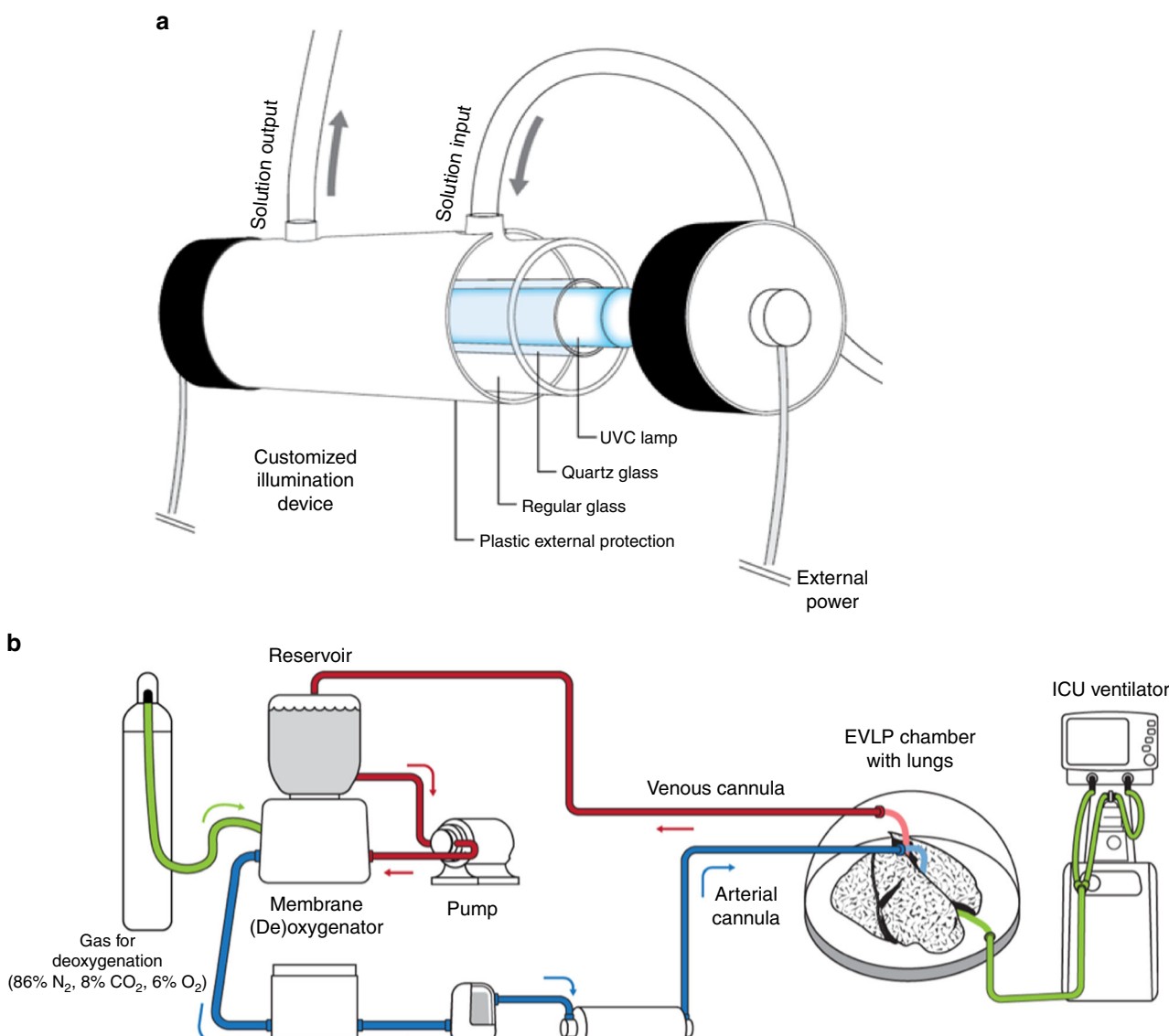

**Fig. 1** The customized illumination device and its usage during ex-vivo lung perfusion (EVLP). **a** The apparatus depicted with a germicidal UVC lamp, which was designed to be used during EVLP allocated in sequence with other EVLP components, in a closed system. Mounted on a cylindric tube, the light source is inserted into a tubular quartz tube, surrounded by an opaque PVC tube, that prevents light from escaping from the illuminated cavity. **b** The EVLP system with the illumination device (irradiator). The lungs are placed into a specific organ chamber. The EVLP circuit is composed of a hard-shell reservoir, a leucocyte filter, a membrane oxygenator/heater and a centrifugal pump. The illumination device, conceived to per part of the EVLP circuit, interpolates the centrifugal pump and the pulmonary artery cannula. During EVLP, the perfusate is treated when illuminated in 360° during its passage

RNA in comparison with the donor's level of viremia. (Supplementary Fig. 2). Viral levels were normalized and expressed as percentage decrease from baseline (1 h EVLP) over time. PDT was the most effective treatment to decrease perfusate HCV RNA levels compared to control lungs: $97.9 \pm 0.71$ vs. $69.5 \pm 0.89\%$ reduction ($p = 0.015$, two-way ANOVA), followed by circuit exchange: $85.8 \pm 2.83$ vs. $46.6 \pm 17.9\%$ reduction ($p = 0.046$, two-way ANOVA). UVC irradiation showed no significant reduction in viral load compared to EVLP alone: $57.6 \pm 6.8$ vs. $54.5 \pm 10.2\%$ reduction ($p = 0.72$, two-way ANOVA). PDT was also the most effective to decrease HCV RNA levels in lung tissue compared to control lungs: $91.0 \pm 0.7$ vs. $75.5 \pm 8.2\%$ reduction ($p = 0.016$, two-way ANOVA), followed by circuit exchange: $84.1 \pm 5.5$ vs. $50.1 \pm 20.1\%$ reduction ($p = 0.0002$, two-way

ANOVA). Similar to the perfusate, the UVC group showed no significant difference when compared to control lungs: $52.0 \pm 5.7$ vs. $53.3 \pm 12.1\%$ reduction ($p = 0.64$, two-way ANOVA) (Fig. 2b). Several key lung functional parameters were evaluated during EVLP: no significant differences were found in graft oxygenation (Delta $PaO_2$), peak airway pressure (PawP) or dynamic compliance (Cdyn) among the groups, suggesting no immediate deleterious effect of UVC or PDT (Supplementary Fig. 3). Delivery of MB during EVLP in a human lung is shown in Supplementary Movie 2.

**Infectivity assessments.** One of the challenges in using quantitative HCV RNA levels as the main efficacy endpoint for LbT

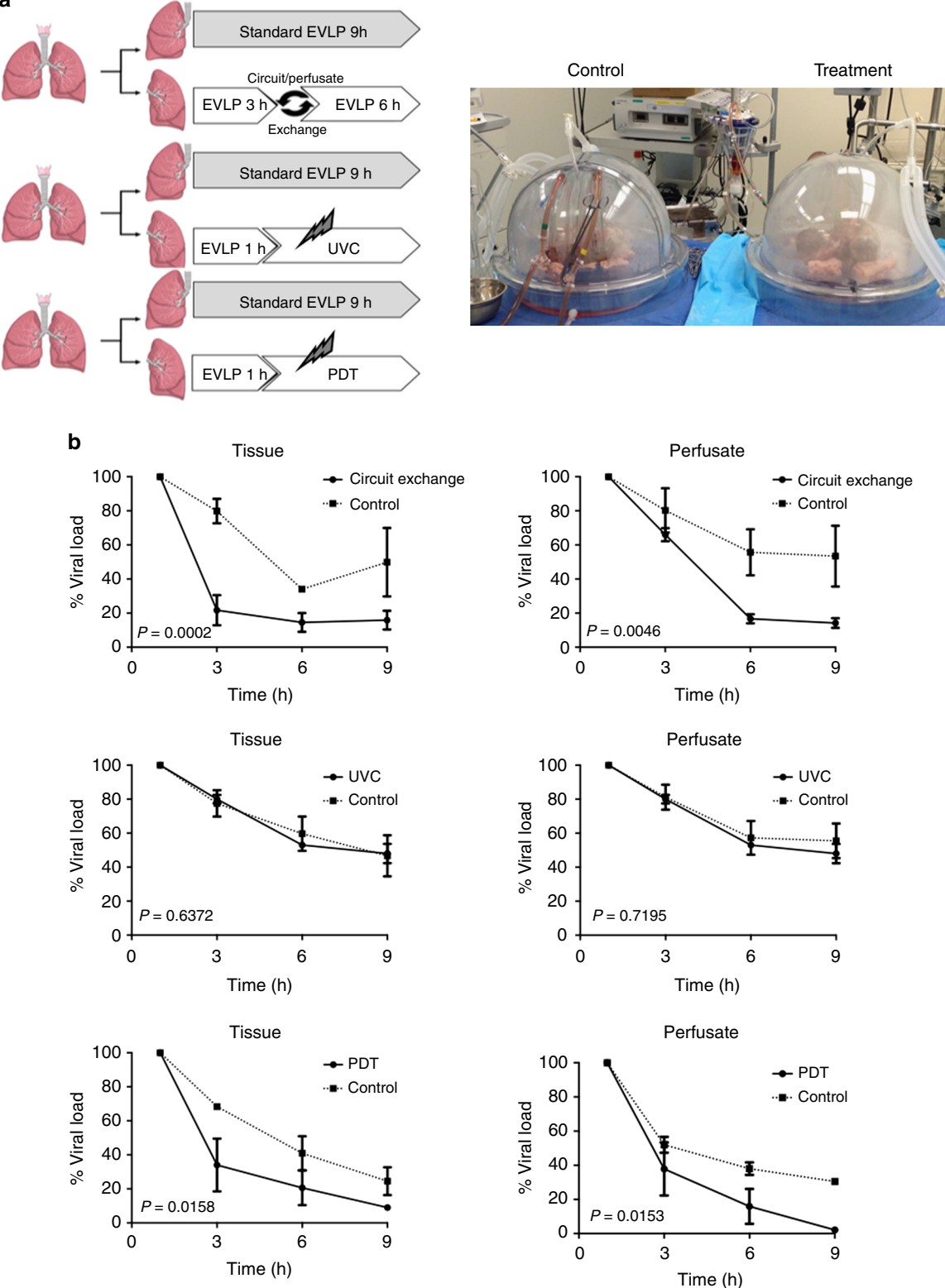

**Fig. 2** Effect of EVLP and light-based therapies (LbT) on HCV RNA levels in HCV NAT + human donor lungs. **a** Paired study design: Lungs from same donor were separated to 2 distinct EVLP systems under different treatment conditions (n = 3, each): standard EVLP (control) vs. treatment (circuit exchange, UVC or PDT). **b** Effect of EVLP and associated treatments towards perfusate and lung tissue HCV levels measured by qPCR during 9 h of treatment. Lung tissue and EVLP perfusate measurement results were normalized for percentage of viral load decrease from baseline and presented as mean ± SEM. The two-way ANOVA statistical test was used for analysis. EVLP: ex vivo lung perfusion

during EVLP is that PDT and UVC may damage and/or fragment virions, rendering them non-infectious. However, damaged virions may still be detected by qPCR. This is particularly relevant because the qPCR assay in the current study targets the 5′-UTR of HCV RNA, which is the most conserved region after LbT[28]. In fact, studies using LbT in blood components have shown that loss of infectivity occurs much earlier than reduction in levels of HCV RNA[22]. This mismatch can be explained by the fact that LbT promotes nucleic acid-crosslinking and oxidative damage, without always causing total fragmentation[29]. We, therefore, hypothesized that our measurement of HCV RNA in the perfusate after EVLP likely underestimated the antiviral effect of LbT, since even detectable virus exposed to LbT may have markedly diminished or absent ability to infect naïve cells. Ideally, infectivity experiments would be replicated with virus from HCV-infected patients after LbT, but HCV is very resistant to growth in in vitro systems[30].

Thus, in order to evaluate the effect of LbT during EVLP on HCV infectivity, we used an HCV cell culture (HCVcc) system based on the Japanese Fulminant Hepatitis-1 clone (JFH-1), which was derived from a genotype 2a HCV isolate. JFH-1 replicates robustly in human hepatoma cell lines that express the viral entry factor CD81, allowing measurement of both HCV RNA levels and HCV infectivity[30]. The ratio of infectious to non-infectious virus is approximately 1 to 1000 for HCV in cell culture[31,32]. We hypothesized that LbT would have a greater effect on the ratio of infectious to non-infectious virus than on the quantitative levels of HCV RNA measured by qPCR. To evaluate this hypothesis, a specifically designed EVLP mini-circuit was developed (Fig. 3a), in which a known amount of JFH-1 virus could be added and photo-irradiated as in the human lung experiments. After being primed with 250 mL of Steen solution, 0.5 focus-forming units/mL (FFU/mL) or 1.5 million copies/mL of JFH-1 HCV was added to the circuit. After the perfusate was warmed to 37 °C, a 180 min treatment was performed in three groups (n = 4 each): (1) Control (no light irradiation), (2) UVC (254 nm, 31 mW/cm²) and (3) PDT (MB at 1, 0.5, 0.1 or 0.01 μmol/L, in different light conditions: dark; 660 nm/20 mW/cm²; ambient room light). During the perfusion time in the circuit, 1.5 mL aliquots were taken and used to spike an Huh 7.5.1 hepatocyte cell culture, which was maintained for 72 h in DMEM supplemented with 10% FBS at 37 °C plus 5% CO₂. The pH was controlled and 30%-90% cell confluence was maintained.

Infectivity was assessed by staining Huh 7.5.1 cells exposed to JFH-1 HCV with DAPI and anti-HCV core antibody to allow counting of clusters of infected cells (FFU). Despite detectable HCV RNA at all time-points after LbT exposure (Fig. 3b), the HCV infectivity of Huh 7.5.1 cells were completely abrogated after 150 min of UVC irradiation in the mini-EVLP perfusate in all experiments (Fig. 3c, d). Similar to the human lung experiments, the PDT group demonstrated even greater efficacy, with no infectivity observed after only 15 min of perfusion under room or red-light exposure. The PDT response was MB dose-dependent. Notably, MB had no antiviral effect when the procedure was performed in the dark (Fig. 4a–c).

**Pre-clinical large animal safety studies using EVLP/LbT treatments**. In order to translate this approach to the clinical setting, we investigated the safety of applying UVC and PDT during EVLP using a pre-clinical large animal lung transplant model[33] comprising male Yorkshire domestic pigs (30–35 kg) as both donors and recipients. After donor lung retrieval, lungs were preserved for 2 h at 4 °C, followed by normothermic EVLP for 6 h (Study design, Fig. 5a). During EVLP, the pig lungs were randomized to three groups (n = 4 each): (1) Control (standard

EVLP); (2) UVC (254 nm; 31 mW/cm²); (3) PDT (1μmol/L MB in the perfusion solution; 660 nm; 20 mW/cm²). Perfusate samples were taken for gas analysis (RAPIDPoint 500 System, Siemens, Munich, Germany). Following EVLP, left lung transplantation was performed and evaluated for 4 h, this being the most critical period to evaluate ischemia-reperfusion injury (post-transplant acute lung injury), followed by right pulmonary artery clamping with total exclusion of the contralateral native lung for isolated graft functional assessment. There were no differences between the 3 groups in physiologic parameters or oxygenation during 6 h of EVLP (Supplementary Fig. 4). More importantly, post-transplant oxygenation function, airway pressures and pulmonary edema did not indicate any signs of measurable lung injury in any of the groups (Fig. 5b–e). Other markers of acute lung injury, including inflammatory and cell-death markers, showed no deleterious effects of UVC or PDT during EVLP (Supplementary Fig. 5) or after transplantation (Figs 5f and 6a–b).

**Clonogenic assays**. Due to the capacity of UVC to alter self-proteins to make them immunogenic and potentially lead to protein repolymerization and photo toxin generation[34], a human fibroblast clonogenic assay was used to assess for any cytotoxicity. Human fibroblasts were seeded out in appropriate dilutions to form colonies in 1–3 weeks. Samples of perfusate (Steen®) solution were irradiated as for the mini-EVLP system, in two different groups (n = 4, each): (1) Control (standard EVLP); (2) UVC (254 nm; 31 mW/cm²), with samples taken at different time points, and then mixed with the cell culture media. Colonies were fixed, stained with crystal violet and counted using a stereomicroscope. We observed no measurable cytotoxic effect of the irradiated perfusion solution (Fig. 7a–b).

**Discussion**

Here we evaluated the therapeutic effect of LbT during organ preservation using normothermic EVLP as a treatment platform. UVC and PDT during EVLP effectively reduced HCV infectivity. Furthermore, studies in a large animal model demonstrated that these therapies, applied during EVLP, have no demonstrable deleterious effects on the lung or to the currently used perfusate solution. These data provide pre-clinical evidence to proceed to clinical studies of LbT during EVLP, especially in an era where available rescue therapies using DAAs have been shown to be efficacious[35]. This therapy could conceivably be translated to other donor organs, such as kidney or heart, from HCV-infected donors. Since no human lung transplants were performed in the present study, we cannot be certain whether transmission would ultimately be prevented. However, this is one of the first attempts in eliminating a viral infection from a human donor organ during preservation, much resembling approaches applied in blood banks in the past. Another recent study demonstrated the viral inactivation efficacy of MB in cells and in ex-vivo mini-pigs kidneys exposed with HCV. In an ex-vivo experiment on contaminated kidney, 1 μM MB was perfused manually or with hypothermic machine perfusion, both resulted in very low or undetectable HCV RNA levels. Compared to the proposed treatment methods in our study, the HCV inactivation mechanism by MB in that study was not investigated, whereas we proposed two photonic treatments, ultraviolet C irradiation and MB-photodynamic therapy[36]. While direct transplantation with an HCV-infected organ coupled with post-transplant DAA therapy of the recipient can potentially be an alternate approach, we believe that organ treatment prior to transplant may have several scientific and clinical advantages: (1) improved societal acceptance of marginal or infected organs/donors, (2)

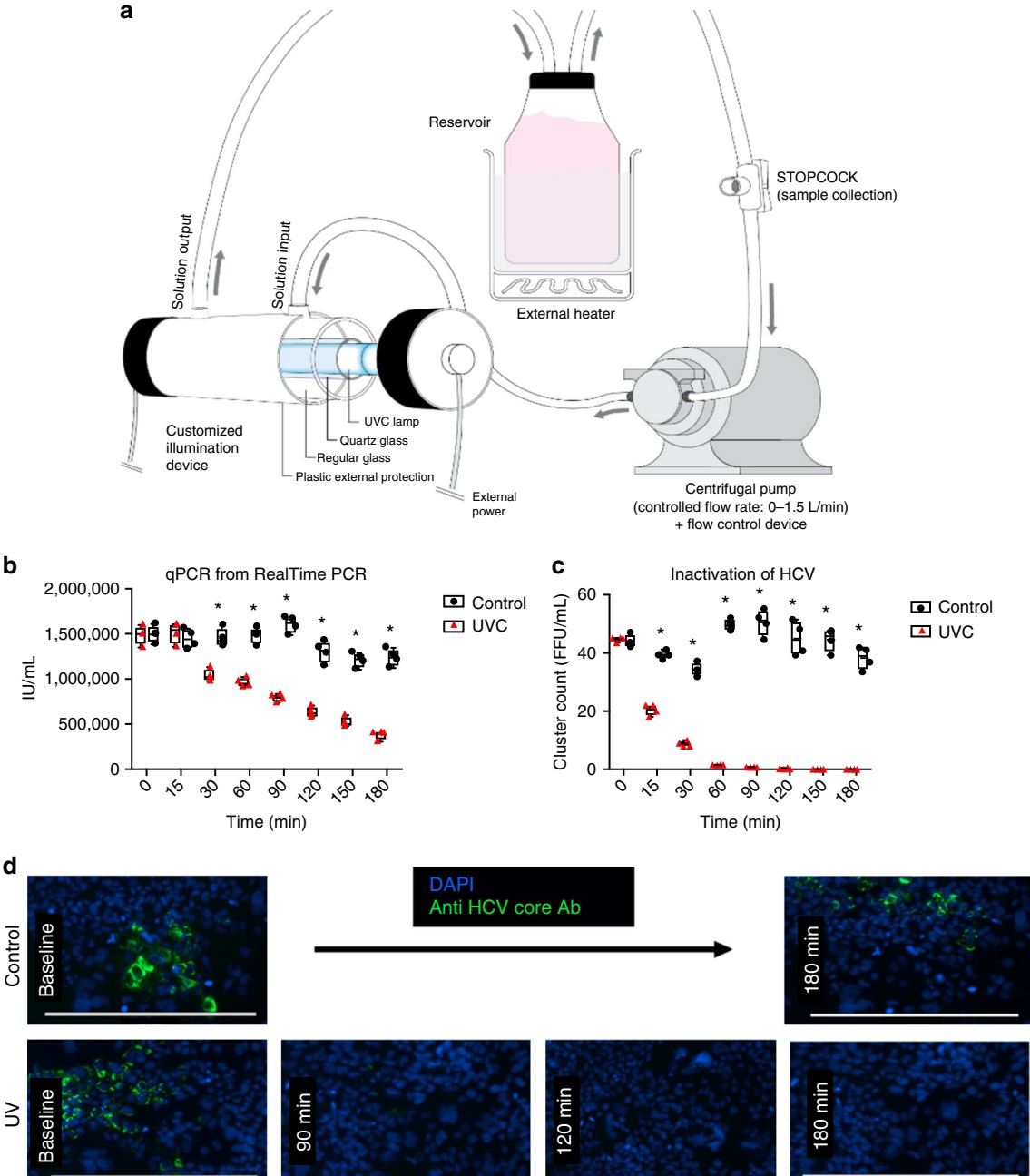

**Fig. 3** Infectivity studies (UVC) **a** Schematic illustration of the mini-EVLP circuit: roller pump, external heater and irradiator device. At the start of perfusion, 1.5 million/ml of HCV genotype 2a surrogate (*JFH*-1) was added to this circuit. **b** After several minutes of UV treatment in the mini-EVLP circuit, perfusate samples were collected and used for HCV quantification by qPCR. After 30 min of UVC irradiation the qPCR counts of HCV significantly decreased in all timepoints ($n = 4$, $p < 0.001$, one-way ANOVA), although virus RNA was still detectable after 180 min of treatment. Black dots represent qPCR results in IU/mL in the control group whereas red triangles represent the qPCR results in the UVC group. Centre line represents mean and bounds of box are standard deviation. **c** Quantification of infectivity loss using cluster counts. Immunofluorescence assessment of HCV infectivity using a hepatocyte cell line (*Huh* 7.5). Cells were double-stained with DAPI and HCV anti-core antibody, then infected hepatocytes clusters were counted. Infectivity rates significantly decreased after 15 min of irradiation ($n = 4$, $p < 0.001$, one-way ANOVA), demonstrating total infectivity loss after 150 min of treatment, in all 4 replicates used. Black dots represent cluster count results in the control group whereas red triangles represent cluster count in the UVC group. Centre line represents mean and bounds of box are standard deviation. **d** Representative immunofluorescence picture of infectivity loss in treated perfusate samples. Scale bar = 400 μm

avoidance of treatment costs associated with DAAs and (3) avoidance of treatment-drug interactions. Furthermore, these ex-vivo therapies may decrease virus loads within the allograft to critically low levels such that very short course DAA treatment immediately post transplantation could lead to successful

prevention of infection. Finally, this EVLP-LbT approach to Hepatitis C treatment opens an opportunity to evaluate prior to transplant ex-vivo treatments of other infectious agents, not only increasing donor pool, but also the safety of transplantation in general.

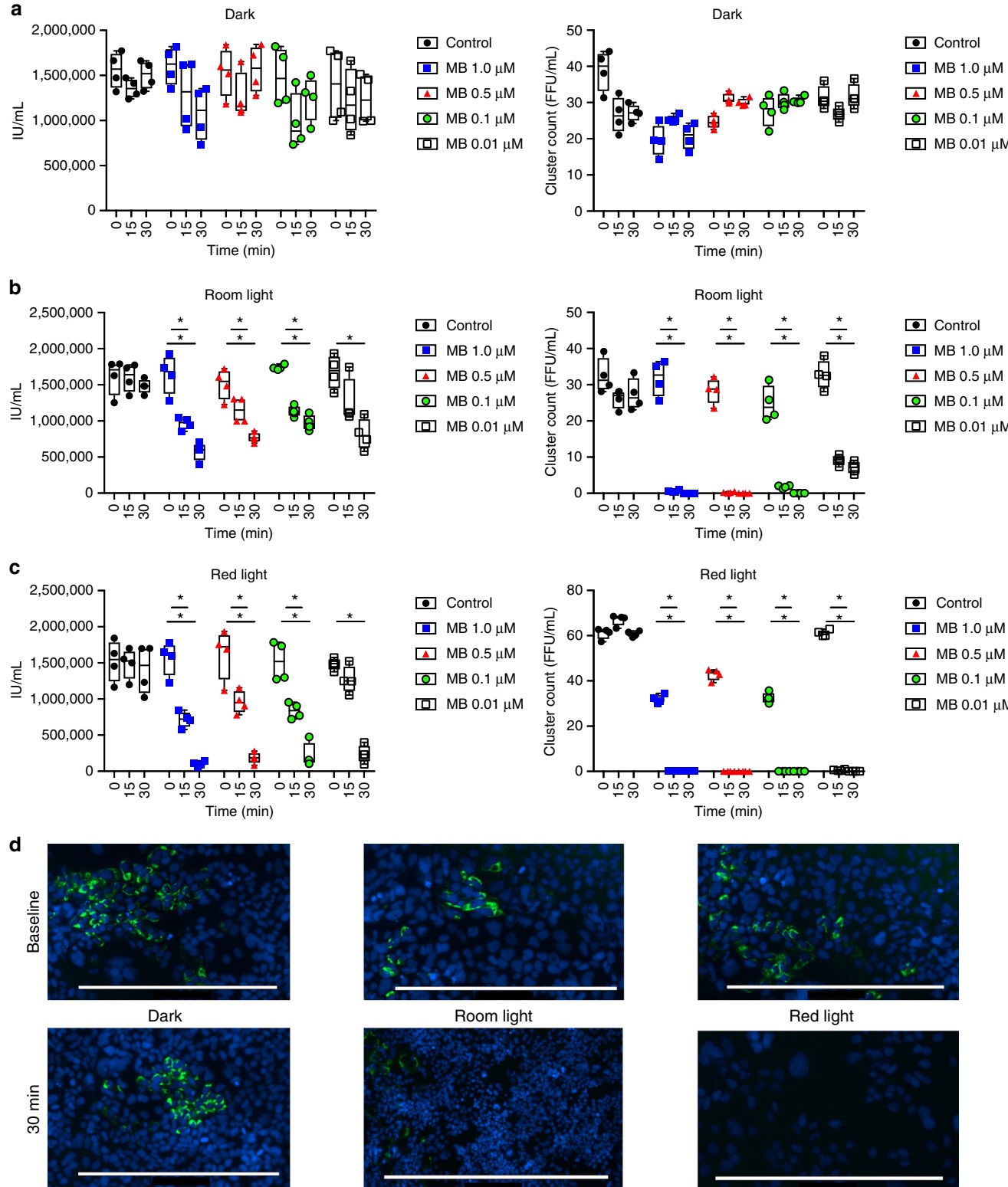

## Methods

**The illumination device.** To treat the perfusion solution during EVLP using a single and ergonomic light source, a customized illumination device was designed and fabricated in collaboration with the University of São Paulo. This is directly connected to the EVLP tubing and so constitutes part of the closed system. It does not affect the perfusion flow or pressure, nor introduces air bubbles after initial priming. The light source is mounted within a quartz tube, surrounded by an opaque PVC tube that prevents light from escaping from the irradiation cavity. The perfusate is thereby irradiated over 360° as it flows through the quartz tube. Being a

closed system, the same fluid liquid passes through the tube several times during EVLP, resulting in a cumulative LbT effect.

**Ex vivo lung perfusion.** EVLP has been established as a clinical tool for LTx[37]. The Toronto technique[1] comprises of perfusion under normothermia, targeting 40% of the donor predicted cardiac output, under protective lung ventilation (7cc/kg, inspired fraction of oxygen of 21%, end expiratory pressure of 5 cm $H_2O$) and the use of an acellular perfusate with increased colloid osmotic pressure (described

**Fig. 4** Infectivity studies (PDT). The perfusion solution in the mini-EVLP circuit was spiked with a set amount of *JFH*-1 viruses and hepatocyte *Huh* 7.5 were used as an infectivity assay ($n = 4$, each group). Methylene Blue (MB) was diluted in different concentrations (1 μM/L, 0.5 μM/L, 0.1 μM/L and 0.01 μM/L) and activated by different light conditions (no light, 660 nm/20 mW/cm² red light and regular room light). A *Huh* 7.5 hepatocyte cell culture was transfected and stained (DAPI and HCV anti-core Ab). Infected hepatocytes clusters were counted. **a** In dark conditions, HCV maintained the infectivity potential, regardless the MB dosage. qPCR counts of HCV were stable during the treatment. **b** Room light exposure promoted virus inactivation in a MB dose-dependent manner ($p < 0.001$ in all scenarios, one-way ANOVA), despite virus RNA still being detected on qPCR. **c** Red light demonstrated the maximal inactivation effectiveness, with significant difference in all scenarios ($p < 0.001$, one-way ANOVA), despite virus RNA still being detected on qPCR. Left panels: Black dots represent qPCR results in IU/mL in the control group whereas qPCR results in different MB concentration are shown in blue (1uM), red (05.uM), green (0.1uM), and white (0.01 uM). (white). Right panels: Black dots represent cluster count results in the control group whereas cluster count results in different MB concentration are shown in blue (1uM), red (05.uM), green (0.1uM), and white (0.01 uM). Centre line represents mean and bounds of box are standard deviation. Centre line represents mean and bounds of box are standard deviation. **d** Representative picture of infectivity loss in treated perfusate samples. Scale bar = 400 μm

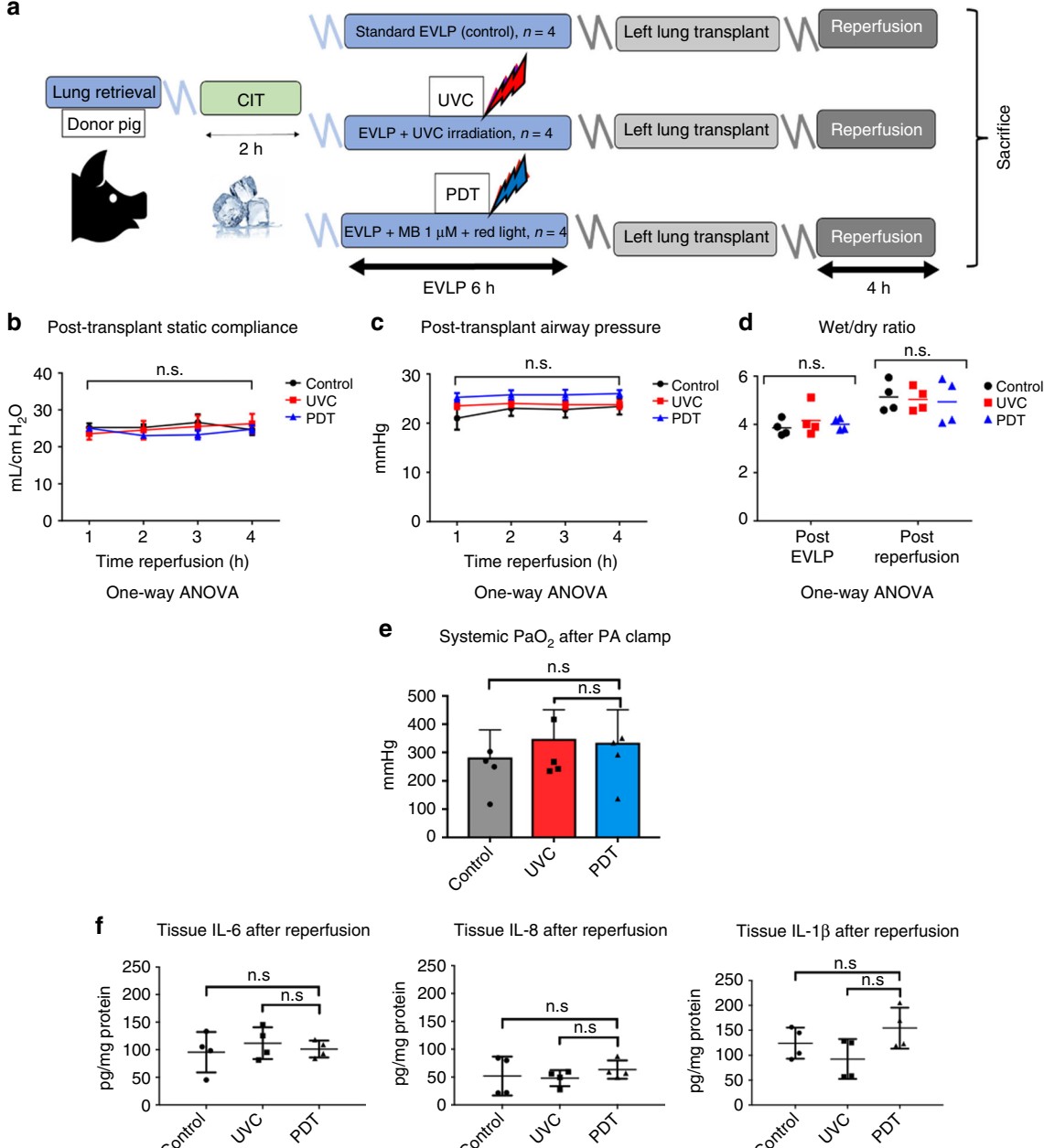

**Fig. 5** Pre-clinical large animal safety studies using EVLP/LbT treatments. **a** Schematic of a pre-clinical EVLP and lung transplantation model, designed to assess potential acute lung injury in donor lungs after LbT applied during EVLP ($n = 4$, each group): (1) Control (standard EVLP technique); (2) UVC (254 nm; 31 mW/cm²); (3) PDT, using 1μmol/L MB diluted in the perfusion solution associated with red light irradiation (660 nm; 20 mW/cm²). **b**–**e** Lung function parameters after left lung transplantation (N.S. after one-way ANOVA statistical analysis). **f** Graft inflammatory cytokine assessment in lung tissue after transplantation (N.S. after one-way ANOVA statistical analysis. Error bar indicate standard deviation)

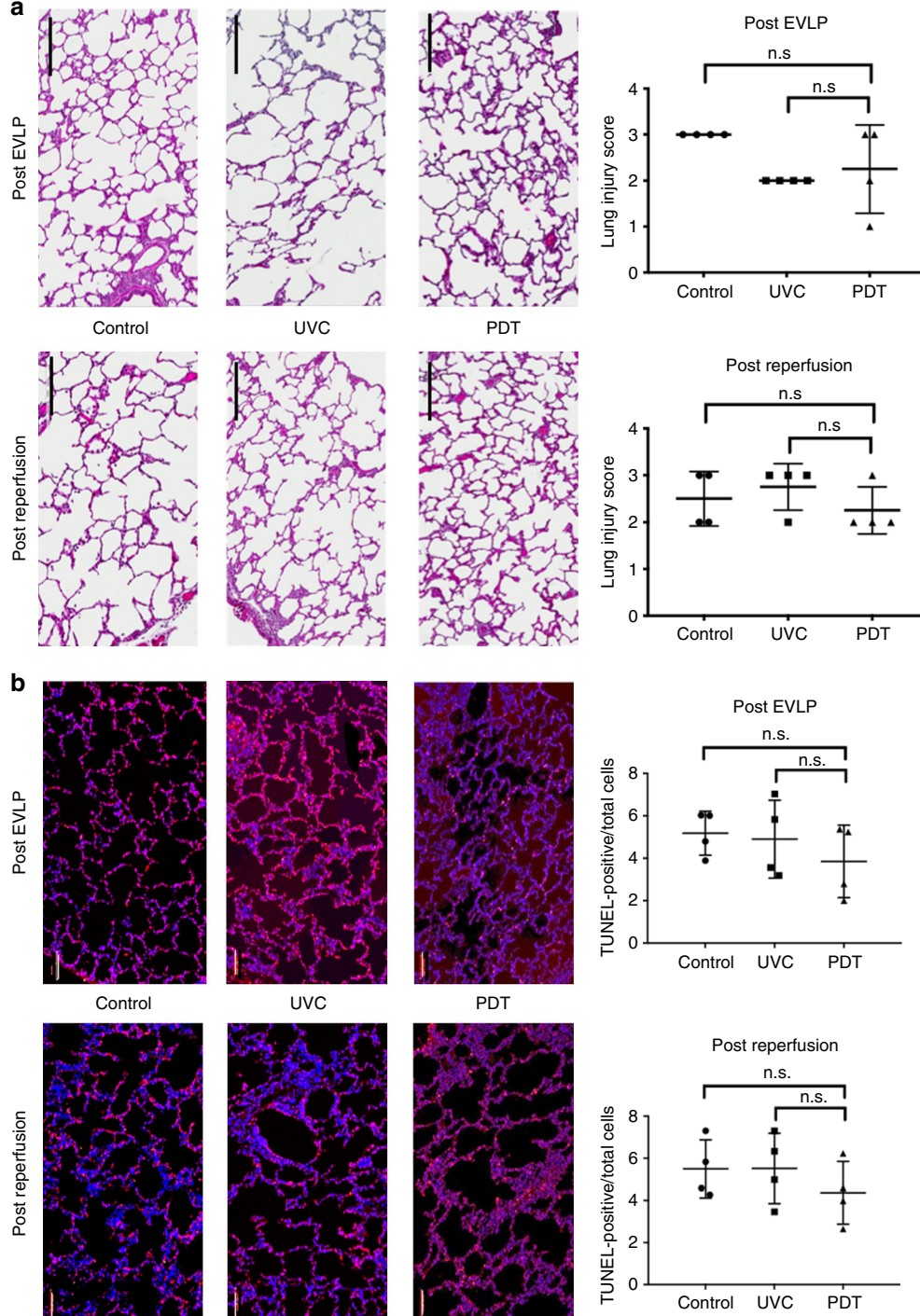

**Fig. 6** Pre-clinical large animal safety studies using EVLP/LbT treatments. **a** Lung injury score after transplantation, scale bar = 100 μm. **b** Cell death assessment (TUNEL) after transplantation, scale bar = 400um (N.S. after one-way ANOVA statistical analysis). MB: methylene blue; CIT: cold ischemia time; PDT: photodynamic therapy; Ultraviolet C (UVC) irradiation; EVLP: ex vivo lung perfusion; LbT: Light based therapy. Error bar indicates standard deviation

below)[37]. After donor lung retrieval, organs are flushed with a low potassium dextran solution and stored inflated at 4 °C. The left atrium (LA) and main pulmonary artery (PA) are cannulated, and an intra-tracheal tube is placed. A second retrograde flush is performed using low potassium dextran solution. The lungs are kept inflated during the entire preparation. After cannulation, the organ is placed inside a customized chamber and the vascular cannulas are connected to the EVLP circuit. The endotracheal tube is connected to an ICU ventilator. The EVLP circuit is comprised of a centrifugal pump, a leucocyte filter, a hollow fiber oxygenator, heat exchanger and hard-shell reservoir. It is primed with 2.0 L of a buffered, low-K + solution with human albumin plus dextran extracellular solution (Steen® solution, XVIVO), 500 mg of methylprednisolone (Solu-Medrol; Sandoz Canada,

Boucherville, Canada), 10,000 IU of unfractionated heparin (Leo Pharma, Thornhill, USA) and 500 mg of Imipenem/Cilastatin, (Primaxin; Merck, Whitehouse Station, NJ)[38].

**HCV NAT + Human Lung EVLP**. Rejected lungs for transplantation in the United States and Canada were used. Lungs received from the United States for research purposes were provided by a third-party company (International Institute for Advancement of Medicine, Edison, USA). Donors were tested for HCV exposure using the nucleic acid test (NAT) and selected if this test was positive. This study was approved by Trillium Gift of Life for donors with research consents and the

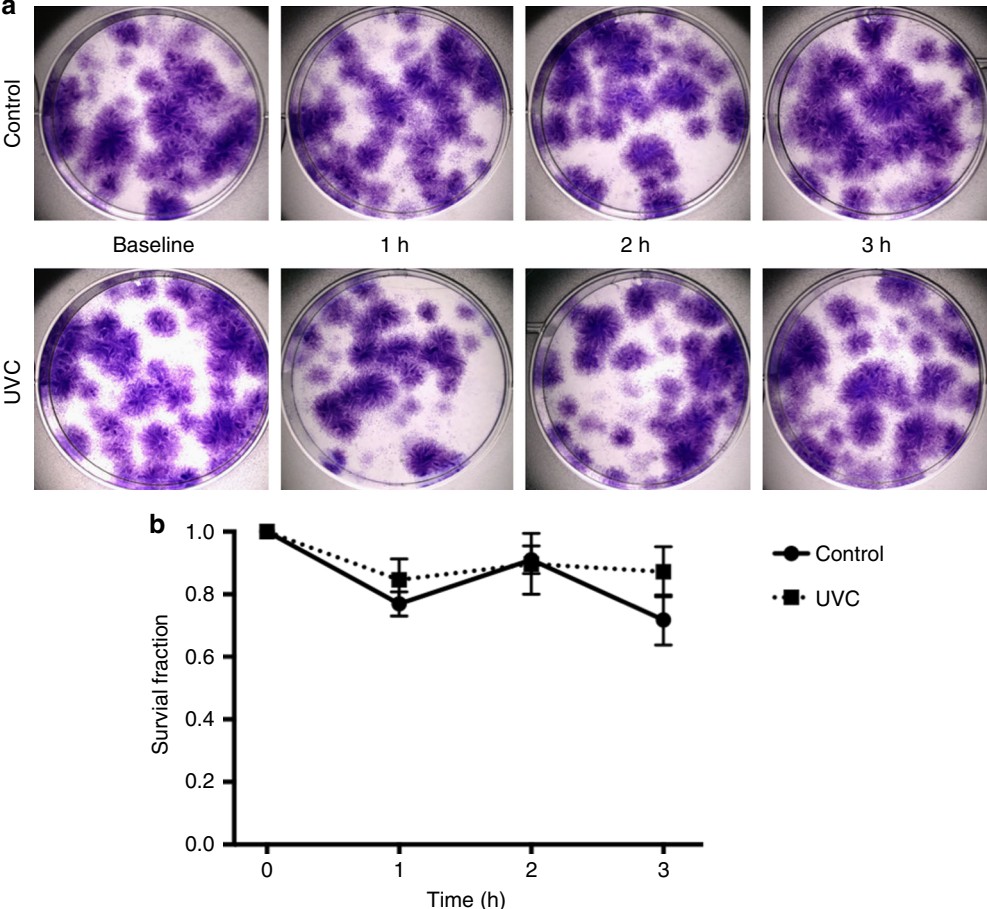

**Fig. 7** Clonogenic cell assay performed in six-well plates, with clones produced by LL 24 ATCC ® CCL-151™ human fibroblasts. Steen was previously irradiated for 3 h while circulating in the mini-circuit and samples were taking hourly. **a** Cells were cultured for 12 days in a mixture of fresh Steen and media (control, upper images) and in a mixture of UVC-treated Steen and media (UVC, bottom images). No cytotoxic effect of the UVC was seen. **b** Survival fraction curves of LL 24 ATCC ® CCL-151™ human fibroblasts ($n = 3$ replicates). The survival curves derived from clonogenic assays experiments and are not significantly different, when comparing untreated controls with cells plated with irradiated Steen solution, after one-way ANOVA statistical analysis

Biosafety office at University Health Network, Toronto. Donor lungs were retrieved and preserved in cold low-potassium dextran solution (Perfadex®, XVIVO Perfusion, CO, USA) on ice until arrival at our institution. The surgical techniques for organ retrieval and EVLP were similar to those used in clinical practice[39]. Upon arrival, lungs were separated, and both lungs were retrograde flushed using Perfadex®. Samples of the perfusate were collected, frozen at −80 °C for later analysis of viral levels using Real Time® HCV PCR. Following the pre-EVLP flush, the lungs were placed in two independent EVLP circuits for 9 h, using the Toronto EVLP technique[1].

**Real Time HCV qPCR**. Samples from donor blood, EVLP perfusate and lung tissue were obtained. All blood samples were separated, and the serum was aliquoted and kept at −80 °C until further testing. Tissue Viral RNA was extracted by the QIAamp viral RNA purification protocol (Qiagen), after homogenization according to the manufacturer's protocol. All samples were genotyped by the Inno-Lipa HCV II test (Innogenetics, Zwijnaarde, Belgium) according to the manufacturer's protocol. HCV RNA is captured by a set of specific, synthetic oligonucleotide target probes. Real-time quantitative RT-PCR was performed, using the M2000 REAL-TIME system (Abbott RealTime HCV, Abbott, IL, USA), and results depicted in International Units Per Mililitre (IU/mL).

**Infectivity assay**. A modified version of *Huh* 7 hepatocyte cell line was used (*Huh* 7.5.1, CVCL_E049, kindly obtained from the National Institutes of Health (NIH), Washington, USA https://web.expasy.org/cellosaurus/CVCL_E049), in DMEM with 10% FBS and at 50% confluence were infected with perfusate from the mini-EVLP or a known quantity of *JFH*1 HCV (virus obtained from Dr. Jake Liang, National Institutes of Health). The cell medium was changed after 3 h and the cells were then maintained at 37ºC for 24 h, after which they were fixed with paraformaldehyde, permeabilized with NP-40 and stained with DAPI and anti-HCV core antibodies at 33% (Anti-Hepatitis C Virus Core antibody, ab58713 Abcam).

Clusters or focus-forming units (FFU) were identified and counted using Image J® software.

**Albumin electrophoresis**. Fresh perfusate was used to prime the miniaturized circuit. After the system was warmed to 37 °C, 1.5 mL aliquots were taken at several time points of light irradiation and analyzed using the Bolt™ 4–12% Bis-Tris Plus Gels assay: 20 µl of 1:400 diluted in H₂O Steen® solution were mixed with 16 µl of SDS sample buffer (0.3 M Tris-HCl, pH 6.8, SDS [6.7% w/v], glycerol [10% v/v], 2-mercaptoethanol [5.3% v/v] and bromophenol blue [0.2% w/v]), boiled for 5 min and stored at 220ºC until SDS PAGE electrophoresis, which was done by separating samples on Bolt™ 4–12% Bis-Tris Plus gels and stained with Simply Blue Safe Stain (Invitrogen, # LC6060).

**Large-animal transplant procedures**. A total of 24 (12 donors and 12 recipients) Yorkshire male pigs (30–35 kg) were used. All animals received humanized care and all protocols were evaluated and approved by the Animal Care Committee, Toronto General Hospital Research Institute, Toronto, following the Canadian Council on Animal Care Certificate of Good Animal Practices Guidelines (https://www.ccac.ca/en/program-features/certificate-of-gap-good-animal-practice.html). Donor pigs were sedated with ketamine (20 mg/kg IM), midazolam (0.3 mg/kg, IM) and atropine (0.04 mg/kg IV) and anesthetized with inhaled isoflurane (3–5%) for peripheral line insertion and intubation. The animals underwent tracheostomy with a 7.5 Fr endotracheal tube and were then placed on a pressure-controlled ventilator at pressure support of 15 cm H₂O, PEEP of 5 cm H₂0, FiO₂ of 0.5 and respiratory rate of 12bpm. Anesthesia was maintained with Propofol (5–8 mg/kg/h IV), and fentanyl citrate (2–20 µg/kg/h IV) or Remifentanil (9–30 µg/kg/hr IV). Cefazolin 20 mg/kg IV and methylprednisolone 500 mg IV were administered prior to skin incision. After baseline parameters and an assessment including blood gas analysis, a median sternotomy was performed and sodium heparin (10,000 IU) was injected systemically. The superior vena cava, inferior vena cava and aorta were clamped, and the lungs were flushed through the pulmonary artery with 60 mL/kg

of low-potassium dextran glucose preservation solution (Perfadex®) at 4 °C from a height of 30 cm above the heart. The heart-lung block was removed in an inflated state and preserved at 4 °C for 2 h. Following lung procurement and cold ischemic time, the lung bloc was placed on EVLP for 6 h, using the Toronto EVLP technique[1].

Following EVLP, the recipient pig was sedated and anesthetized using the same technique as for the donor pig. The transplant consisted of a left thoracotomy through the fifth intercostal space. The pulmonary hilum was dissected, and the left azygous vein was ligated. The inferior pulmonary ligament was divided. Both the right and left main pulmonary arteries were carefully dissected. After administering heparin, the left pneumonectomy was completed. The bronchial anastomosis was performed first, with a continuous 4–0 synthetic, monofilament, non-absorbable polypropylene suture interrupted in two places. The PA anastomosis was then performed, with a continuous 5–0 synthetic, monofilament, non-absorbable polypropylene suture, interrupted in two places. Left atrial anastomosis was then performed with an everting continuous 5–0 synthetic, monofilament, non-absorbable polypropylene suture interrupted in two places, after which the transplanted lung was re-inflated with recruitment to a pressure of 25 cm $H_2O$ and ventilated for 4 h, under the following ventilatory parameters: pressure-controlled (PC) ventilation at pressure support of 15 cmH$_2$O, PEEP of 5 cmH$_2$0, FiO$_2$ of 0.5 and respiratory rate of 15/min targeting tidal volumes of 6–8 ml/kg. The right pulmonary artery was clamped 4 h after reperfusion to assess the function of the (left) transplanted lung alone. After 4 h of functional assessment, the animals were sacrificed by exsanguination.

**Lung edema**. Lung tissue biopsy samples were collected after 6 h of EVLP and 4 h of reperfusion (post-transplant) and were used to assess lung edema (wet-dry ratio) and histological lung injury. For wet-dry ratio, lung tissues were weighed fresh and placed in an oven at 85 °C for 72 h to dry and re-weighed to calculate the level of lung edema.

**Acute lung injury assessments**. Lung tissue samples were embedded in paraffin after fixation in 10% buffered formalin for 24 h, followed by 5 μm thick sectioning and H&E staining. Mid-sagittal sections were assessed in a blinded fashion (Y.W.). We evaluated interstitial edema, intra-alveolar edema, hemorrhage, cell infiltration and hyaline membrane formation. The severity of these findings was graded in a four- point scale as follows: 0, absent; 1, mild; 2, moderate and 3, severe.

**Cytokine assays**. Lung tissues and perfusate samples stored at −80 °C were evaluated for inflammatory response using enzyme-linked immunosorbent assay (ELISA) for porcine IL-6, IL-8, and IL-1β, in tissue samples were homogenates. Supernatants and perfusates were assayed using commercial ELISA kits for porcine interleukin IL-1β (DY6226, IL-1 beta Pig ELISA Kit, R&D Systems, Minneapolis, MN), IL-6 (P6000B, Porcine IL-6 Immunoassay, R&D Systems, Minneapolis, MN), and IL-8 (P8000, IL-8 beta Pig ELISA Kit, R&D Systems) according to the man-ufacturers' instructions. The final concentration was calculated by converting the optical density (OD) readings against a standard curve.

**Apoptosis assessment**. Lung tissue biopsy samples were collected after 6 h of EVLP and 4 h of reperfusion (post-transplant). These were stained with deox-ynucleotide transferase-mediated deoxy uridine triphosphate nick-end labeling (TUNEL) (In Situ Cell Death Detection Kit, POD; Roche Diagnostics GmbH, Mannheim, Germany). All TUNEL-stained slides were scanned using a whole-slide fluorescence scanner (Axio Scan.Z1, Carl Zeiss Microscopy GmbH, Jena, Germany). TUNEL-positive cells were counted using commercial image-analysis software (HALO™ Image Analysis Software, Pelkin Elmer, Waltham, MA) and expressed as a percentage of total cells.

**In vitro clonogenic cells assay**. The clonogenic assay of cells is considered the gold standard to determine cell reproductive death after treatment with ionizing radiation, but can also be used to determine the effectiveness of other cytotoxic agents, such as photoproducts after ionizing radiation or PDT[40]. In this study the cell handling, assay setup, fixation, staining and colony counting technique were performed following published reference guidelines[40]. Human fibroblasts (LL24) were purchased from American Type Culture Collection (ATCC) and cultured in McCoy's media supplemented with 15% of fetal bovine serum. Two-hundred cells were seeded in 6-well Petri dishes and exposed to Steen® solution, previously irradiated in the EVLP circuit with UVC or not, for 24 h. The media was then replaced, and the cells were kept in the incubator at 37 °C and 5% $CO_2$ for 12 days. Colonies were then fixed with formalin, stained with crystal violet, imaged with a stereomicroscope and counted using Image J® software. The plating efficiency and the survival fractions were determined as following protocols published by Franken and collaborators[40].

**Statistical analysis**. All results are expressed as mean ± standard error of the mean. Unpaired Student's *t* test and Mann–Whitney nonparametric test were used for comparisons between groups. For comparisons between groups at all time points, two-way repeated-measures analysis of variance (ANOVA) was used. Significance was considered as $p \leq 0.05$.

**Reporting Summary**. Further information on experimental design is available in the Nature Research Reporting Summary linked to this Article.

## Data availability
The data sets generated during and/or analyzed during the current study are available from the corresponding authors on reasonable request.

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

## Acknowledgements

This work was supported by the Canadian Institutes of Health Research (CIHR Project Grant), Medicine-by-Design Cycle 1 Team Projects Award (# C1TPA-2016-07), Toronto General & Western Hospital Foundation (Grant # 1013612) and XVIVO Perfusion. Financial support was also provided by the Brazilian agencies São Paulo Research Foundation (FAPESP CEPOF 2013/07276–1, INCT 2014/50857–8) and National Council for Scientific and Technological Development (CNPq 465360/2014–9). We thank Paul Chartrand (Latner Thoracic Surgery Laboratories) for supplies and logistics management, Natalia Mayumi Inada, José Dirceu Vollet-Filho, Mariana Carreira Geralde and Ilaiáli Souza Leite (IFSC-USP, SP, Brazil) for technical help, and Daisuke Nakajima, Ashish Patel and Hemant Gohkale (Latner Thoracic Surgery Laboratories) for assisting during EVLP and transplantation.

## Author contributions

Conception and design: M.G., M.P., V.S.B., C.K., J.J.F., A.H., and M.Cypel. Human Lung Experiments: M.G., Y.W., C.S., M.P., A.A., R.Q., M.Chen., R.V.P.R., K.R., and G.M. Animal Experiment: M.G., Y.W., C.S., A.A., R.Q., M.Chen., R.V.P.R., K.R., and G.M. Analysis and interpretation: M.G., Y.W., M.P., A.G., A.H., L.P., V.C., T.W., M.L., S.K., and M.Cypel. Drafting the manuscript for important intellectual content, M.G., J.J.F., A.H., V.S.B., C.K., M.L., B.C.W., and M.Cypel.

## Additional information

**Competing interests:** M.Cypel, T.W., S.K. and M.L. are founders of XOR Labs Toronto and M.Cypel, T.W. and S.K are consultants for Lung Bioengineering. J.J.F is consultant for AbbVie, Gilead Sciences, Merck and ContraVir. The remaining authors declare no competing interests.

