## [Peer Review File · Nature Communications]

Editorial Note: This manuscript has been previously reviewed at another journal that is not operating a transparent peer review scheme. This document only contains reviewer comments and rebuttal letters for versions considered at Nature Communications .

Reviewers' Comments:

Reviewer #1:

Remarks to the Author:

The manuscript by Galasso et al. describes new methods for the inactivation of hepatitis C virus (HCV) in donor lungs. As HCV-positive organs are not considered for transplantation and as there is a shortage of donor organs, these methods could potentially lead to an increased donor organ pool. One could argue that instead of inactivating the virus in HCV-positive organs, the new direct-acting antivirals (DAAs) could be used to treat patients receiving HCV-positive donor organs. However, DAAs are not only expensive, but they are contraindicated in some patient groups. Thus, this reviewer thinks that the methods developed by Galasso et al. are interesting and could be a promising approach besides the use of DAA to include HCV-positive donors into the organ pool. Most of my previous comments have been addressed. I still believe that next generation sequencing would be beneficial for the study and would complement the infectivity data. A 100-fold reduction (down to 1% of the input) is good, but with a high viral of 10^6 that would mean still 10^4 RNA copies. This should be at least implemented in future clinical trials. NGS is nowadays widely commercial available even with bioinformatic support.

Reviewer #2:

Remarks to the Author:

After reviewing the revised manuscript I can repeat my initial statement that this work is very well done and of high clinical interest and relevancy. The amendments made, following the reviewer suggestions, do strengthen the paper further.

Reviewer #3:

Remarks to the Author:

This paper was previously reviewed and noted by reviewers and editors to be scientifically very well done and methodically demonstrate substantial inactivation of HCV from donor lungs. The major comment from all reviewers remains the question of whether or not ALL virus has been inactivated from lung tissue/lymphatics as likely if any infectious virus remains, there will be transmission to the recipient. Authors note, this question can only be addressed in a clinical trial, which is planned. All minor comments have been addressed and discussion added around the recently published paper by Helfritz et al. The questions of cost effectiveness remain, but are beyond the scope of this paper. Authors cite a Canadian price of EVLP, but a private insurer price of DAA. Although not publicly disclosed, the Canadian price of DAA for those with public coverage, which is the majority, is speculated to be 15-20k. thus if DAA still needed after EVLP cost will be double. Short course therapy likely only to be effective if used pre-emptively, not as therapy after transmission. Considerations to think about when designing/implementing clinical trial.

REVIEWERS' COMMENTS:

Reviewer #1 (Remarks to the Author):

The manuscript by Galasso et al. describes new methods for the inactivation of hepatitis C virus (HCV) in donor lungs. As HCV-positive organs are not considered for transplantation and as there is a shortage of donor organs, these methods could potentially lead to an increased donor organ pool. One could argue that instead of inactivating the virus in HCV-positive organs, the new direct-acting antivirals (DAAs) could be used to treat patients receiving HCV-positive donor organs. However, DAAs are not only expensive, but they are contraindicated in some patient groups. Thus, this reviewer thinks that the methods developed by Galasso et al. are interesting and could be a promising approach besides the use of DAA to include HCV-positive donors into the organ pool.

Most of my previous comments have been addressed. I still believe that next generation sequencing would be beneficial for the study and would complement the infectivity data. A 100-fold reduction (down to 1% of the input) is good, but with a high viral load of 10^6 that would mean still 10^4 RNA copies. This should be at least implemented in future clinical trials. NGS is nowadays widely commercially available even with bioinformatic support.

Thank you for your positive comments. We will plan to perform NGS in our upcoming clinical trials using UVC and PDT in ex vivo lung perfusion as suggested.

Reviewer #2 (Remarks to the Author):

After reviewing the revised manuscript I can repeat my initial statement that this work is very well done and of high clinical interest and relevancy. The amendments made, following the reviewer suggestions, do strengthen the paper further.

Thank you very much

Reviewer #3 (Remarks to the Author):

This paper was previously reviewed and noted by reviewers and editors to be scientifically very well done and methodically demonstrate substantial inactivation of HCV from donor lungs. The major comment from all reviewers remains the question of whether or not ALL virus has been inactivated from lung tissue/lymphatics as likely if any infectious virus remains, there will be transmission to the recipient. Authors note, this question can only be addressed in a clinical trial, which is planned. All minor comments have been addressed and discussion added around the recently published paper by Helfritz et al. The questions of cost effectiveness remain, but are beyond the scope of this paper. Authors cite a Canadian price of EVLP, but a private insurer price of DAA. Although not publicly disclosed, the Canadian price of DAA for those with public coverage, which is the majority, is speculated to be 15-20k. thus if DAA still needed after EVLP cost will be double. Short course therapy likely only to be effective if used pre-emptively, not as therapy after transmission. Considerations to think about when designing/implementing clinical trial.

Thank you for your very thoughtful comments. We agree the current paper cannot definitively state if no infectious virus is still present in some compartments of the organs. In regards to your trial design suggestions we agree and in fact we are designing a pre-emptive trial with ultra-short therapy with or without EVLP/UVC.